# Sequential Models for Endoluminal Image Classification

**DOI:** 10.3390/diagnostics12020501

**Published:** 2022-02-15

**Authors:** Joana Reuss, Guillem Pascual, Hagen Wenzek, Santi Seguí

**Affiliations:** 1Department of Mathematics and Computer Science, Universitat de Barcelona, 08007 Barcelona, Spain; guillem.pascual@ub.edu (G.P.); santi.segui@ub.edu (S.S.); 2Chair of Remote Sensing Technology, Technical University of Munich, 80333 Munich, Germany; 3CorporateHealth International ApS, 5230 Odense, Denmark; hagen@corphealth.co

**Keywords:** polyp detection, wireless capsule endoscopy (WCE), endoluminal image classification, neural networks, sequential models

## Abstract

Wireless Capsule Endoscopy (WCE) is a procedure to examine the human digestive system for potential mucosal polyps, tumours, or bleedings using an encapsulated camera. This work focuses on polyp detection within WCE videos through Machine Learning. When using Machine Learning in the medical field, scarce and unbalanced datasets often make it hard to receive a satisfying performance. We claim that using Sequential Models in order to take the temporal nature of the data into account improves the performance of previous approaches. Thus, we present a bidirectional Long Short-Term Memory Network (BLSTM), a sequential network that is particularly designed for temporal data. We find the BLSTM Network outperforms non-sequential architectures and other previous models, receiving a final Area under the Curve of 93.83%. Experiments show that our method of extracting spatial and temporal features yields better performance and could be a possible method to decrease the time needed by physicians to analyse the video material.

## 1. Introduction

### 1.1. Motivation and Contribution

Colorectal cancer is the third most frequent type of cancer, diagnosed in both men and women [1]. However, it is also the cancer type where an early treatment has the highest chances of success. As such, detecting early warning signs and connected diseases, such as internal bleeding or polyps [2], is greatly important. The most common method for this purpose is endoscopy. In this diagnostic procedure, an endoscope (a long tube with a camera attached to it) gets inserted into the patient’s body to allow physicians to inspect the gastrointestinal (GI) system. This often causes patient discomfort and requires substantial time, medical staff, and infrastructure. Therefore, an alternative technique called Wireless Capsule Endoscopy (WCE) has been developed by Swain et al. [3]. WCE is a procedure in which patients swallow a pill-sized capsule that contains a small camera. While moving through the patient’s GI system, the camera takes up to 400,000 images [4] to capture abnormalities. While this alleviates the discomfort patients experience during endoscopy, it comes at the expense of additional post-analysis requirements. After the device has transmitted the captured images to an external receiver, they must be analysed by physicians or other medical experts. Because of the large number of images taken during the procedure, manual analysis is time- and labour-intensive. Thus, computer-aided detection methods have received increased attention aiming to identify GI abnormalities [5]. Most of these methods focus on the detection of polyps or tumours. These systems usually serve as decision support systems to point the physician at the most conspicuous images that have a high chance of showing a polyp [2].

Figure 1 shows nine random images from the same procedure. It can be observed that the images captured during WCE can differ drastically from one another. Figure 2 shows two sequences that have been obtained from different patients. Both sequences (partly) contain frames showing a polyp. As can be seen, because of the camera’s movement, the frames within a sequence can vary significantly.

The numerous images taken during WCE result in a final low frame rate video of multiple hours. Convolutional Neural Networks [6] (CNNs) only consider spatial information, meaning a single frame of the video, and therefore do not consider any temporal context. Sequential Models, such as some type of Recurrent Neural Network [7] (RNN), could exploit that information. This puts forward whether Sequential Models can improve the performance of a CNN in WCE polyp detection. In this work, we compare the performance of using Sequential Models for fine-tuning with using solely networks that extract only spatial information. We show that, by taking into account the temporal nature of the data, the detection of polyps in WCE videos can be improved.

### 1.2. Related Work

Several approaches have been proposed to detect intestinal abnormalities in WCE images and videos within the past decade. Among others, they targeted the recognition of bleeding, polyps, tumours, and motility disorders. Before 2015, polyp detection algorithms centred on conventional, typically handcrafted Machine Learning methods. They usually targeted one of three main feature areas to detect abnormalities: colour, shape, or texture. Li and Meng [8] focused on the latter and used wavelet transformations as well uniform local binary patterns with Support Vector Machines as the final classifier. Yu et al. [9] were among the first to investigate Deep Learning techniques. They proposed an architecture named HCNN-NELM, which uses a Convolutional Neural Network as a feature extractor and a cascaded Extreme Learning Machine (ELM) classifier. ELMs are known to result in superior performance than SVM and the fully connected layer of a CNN [9].

Yuan and Meng [10] used a novel method named stacked sparse autoencoder with image manifold constraint (short SSAEIM). This led to two subsequent publications featuring different approaches. The first system, called RIIS-DenseNet, consisted of a DenseNet using two loss functions and could outperform the previous results [11]. The idea was based on the argument that high intra-class variability and object rotation significantly hinder the performance of prior approaches. In a subsequent publication, they developed a slightly different system. This one not only aimed at overcoming high intra-class variability and low inter-class variance but also data imbalance [12]. In order to do so, the authors proposed a so-called DenseNet-UDCS, which uses a DenseNet (like in the previous publications) together with an “unbalanced discriminant (UD)” and a “category sensitive (CS)” loss. This helped to calculate discriminative and appropriate features.

Furthermore, Guo and Yuan [13] introduced a system named Triple ANet using Adaptive Dense Block (ADB) and Abnormal-aware Attention Module (AAM). This helped to capture correlations and highlight informative areas in the images [2]. Additionally, they introduced a loss named Angular Contrastive Loss (AC Loss) to help deal with high intra-class variabilities and low inter-class variances.

Laiz et al. [2] argued that especially the diversity of polyp appearance and the highly imbalanced and scarce data make this research area challenging. They aimed to improve the feature extraction in the case of small datasets using a Triplet loss function. A Triplet loss represents images from the same category by similar embedding vectors, whereas images from different categories are represented by non-similar vectors [2].

Finally, to the best of our knowledge, Mohammed et al. [14] proposed the only currently existing work of using RNNs for colon disease detection. They argued that frame level labels are rarely available in the clinical context and therefore proposed a network named PS-DeVCEM that learns multi-label classification on the frame level. It uses a CNN and a residual Long Short-Term Memory Network [15], short LSTM, that extract the spatial and temporal features, respectively. The additionally implemented attention mechanism and self-supervision methods to allow the system to minimise within-video similarities between positive and negative feature frames.

## 2. Materials and Methods

### 2.1. Dataset

We used retrospective Wireless Capsule Endoscopy (Medtronic PillCam COLON2) data from 110 patients were conducted on behalf of the NHS Highland Raigmore Hospital in Inverness. All patients from this study come from referrals for symptoms or were on surveillance lists within the Highlands and Islands area of Scotland and had a positive Faecal Immunochemical Test (FIT). Patients with hemorrhoidal bleeding were excluded from this study, as well as all that met standard exclusion criteria for Colon Capsule Endoscopy (CCE), especially wearers of heart rate monitors and patients with diabetes.

Each video was tagged according to whether the experts had identified a polyp in an image or not. It was analysed first by a pre-reading nurse, and then a final report was obtained by a medical doctor. In total, the analyses were conducted by eight different pre-readers and two medical doctors. Pre-readers were endoscopy nurses with at least three months of experience with CCE. They received formal CCE training and conducted 5 to 20 video analyses per week. Daily, they followed a standard operating procedure to ensure that each video was analysed consistently, repeatable, and documented according to common standards. Two medical doctors obtain a final report (one gastroenterologist and one internal medicine) with two and five years of CCE experience, respectively.

All images are coloured and have a resolution of 256×256 which leads us to a dimension of [H,W,C]=[256,256,3] with H,W representing height and width and *C* the number of colour channels. We load the data in sequences of 9 images for the Sequential Models, resulting in one sequence having the format [Seq_len,H,W,C]=[9,256,256,3].

Our network is aimed to be used on top of a pre-trained model, such as the system from Laiz et al. [2], and aid in classifying hard examples. The pre-trained system should take care of simple and easy images, while ours would handle those that prove hard to classify. Thus, we need a specialised version of their dataset, pre-filtered with the samples that meet the above conditions. All positive samples (i.e., all polyps) were selected, independently of their difficulty, to ensure that our system does not lose sensitivity nor fails in classifying obvious positives. The negatives, however, were filtered by considering only the images that were hard to classify with Laiz et al. [2] model. Finally, to receive a binary class distribution, the images were tagged by the network as follows:$$\hat{y}\left(x\right)=\left\{\begin{array}{cc} 0, & \mathrm{normal}\\ 1, & \mathrm{polyp}, \end{array}\right..$$

The dataset comprises 110 videos from 110 patients, where each video contains between 49 and 54 sequences, of which nine images represent each. As represented in Table 1, the dataset is partitioned in five different folds, each with its independent train and test sets. Our splits follow the same partitions as proposed in Laiz et al. [2], to allow for fair comparisons. Further details about the selection criteria for the folds and the evaluation strategy are presented in Section 2.4.2.

Table 1 also shows that only 4.22% of all images have polyps, resulting in a high variance of polyps per sequence. About 94.5% of sequences do not show any polyps at all. Moreover, sequences that exclusively contain images with polyps are relatively rare and cover only 1.9% of all sequences. One way to deal with the present class imbalance is to balance out the batches for the training data. Meaning, we have one sequence with polyps for every four sequences without. In that way, one training epoch is defined as the number of iterations until all samples from the positive class, meaning all sequences with at least one polyp, have been returned. A similar procedure has been conducted in Laiz et al. [2].

### 2.2. Outcomes

The primary outcome is a more robust model for polyp detection, with an overall greater rate of detection and reduced false positive count. Secondary outcomes include the certainty that averaging the temporal nature of WCE videos in Machine Learning models helps to obtain better results, producing more accurate models than previous systems that did not use this approach.

### 2.3. Methodology

LSTMs [16] are perfectly built to process sequential information since time can be taken into account as an additional dimension. Hence, they not only consider the present input but also connect past information to the present time step. This is made possible through a feedback loop. The past information is stored in form of a hidden state vector $h_{t-1}$. These hidden states represent the previous inputs and can flow from one step to the next. They outperform regular RNNs by overcoming the problem of short-term memory, making them better at learning long-term dependencies, struggling less at predicting a time step at the end of a more extended sequence. To that effect, an LSTM’s hidden layers contain cells which have three so-called gates: an input, an output, and a forget gate. The network can use these gates to filter what information to add or remove from the hidden state and the current input. Therefore, they are more likely to retain only relevant information. The forget gate decides what information of the cell state $C_{t-1}$ to keep, through a sigmoid activation, whereas the input gate controls what new input information will be fed into the cell state with a combination of sigmoid and hyperbolic tangent activations. The cell state vector contains long-term dependencies. It is used by the backpropagation algorithm to keep the gradients from vanishing [17]. The output gate controls the information flow that leaves the cell to compute the new hidden state with another sigmoid activation.

Regular LSTMs are unidirectional, meaning they only use outputs of previous time steps within the sequence as inputs for the present time step, but not the other way around. A way of improving the recognition rate is to implement a memory that works in both directions, specifically, by implementing an LSTM that can use the output of the present step to update previous time steps. This kind of network is called bidirectional LSTM [18] (BLSTM). For an input sequence $x_{1},\ldots x_{N}$, bidirectional LSTMs process the input in the forward direction, starting with $x_{1}$ and in the backward direction starting with $x_{N}$. Hence, instead of only training one LSTM, the BLSTM simultaneously trains two LSTMs. Both layers’ results are then concatenated to deliver the respective output. A bidirectional LSTM is known to perform better than the unidirectional version, since it has more contextual information available. BLSTMs can exploit information from later time steps, whereas LSTMs cannot dertake into account later steps in the sequence to classify previous steps. Let us assume we have a sequence of polyps. The last two frames are easy to classify, whereas the first ones are hard to detect (for example, because they are not clearly visible because of the camera’s movement). A BLSTM now processes both the information from the hard-to-classify images at the start and the easy frames at the end of the sequence and concatenates all. As a result, the network has more information available and is more likely to make the correct prediction. Hence, BLSTMs are known to perform better for cases where later time steps in the sequence yield necessary information or patterns that can influence the classification of the current time step.

Therefore, they are perfectly suitable for data received through WCE, and we assume them to outperform the SSL and SSL CNN architecture used in this paper.

In this work, we propose to use the self-supervised network (SSL) from Pascual et al. [19], which was trained on the original, non-filtered dataset, as a low-level feature extractor as the backbone of our model. We then add a CNN, an LSTM, or a BLSTM respectively on top, where the latter two extract the high-level temporal information. All three layers are configured with 2048 hidden units. Particularly, the BLSTM layer uses 1024 units for the forward pass and 1024 for the backward one. The CNN uses ReLU activations [20], while both the LSTM and the BLSTM use the activations described above. We test the pre-trained SSL architecture on our dataset. Figure 3 shows the final architecture of the SSL BLSTM network. We have found that dropout layers are not needed to prevent overfitting. Similarly, we do not perform any kind of L1 or L2 regularization. The pretrained network where the embeddings come from, though, was L2-regularized.

The baseline model and overall architecture work the same way for all three proposed networks. We use a ResNet [21] with 50 layers. The ResNet receives the initial flattened sequence of images and extracts the local features. Following their proposed architecture, we then perform average pooling before reshaping the output of the ResNet by again undoing the previous flattening in order to feed it into the bidirectional LSTM. We zero out the classifier’s gradient to freeze the pre-trained ResNet parameters and no longer update them. By then feeding the output (which holds the extracted features) to the BLSTM, we can reduce temporal variations [22]. Its output is then passed to a fully connected (FC) layer. Subsequently, binary class discrimination is performed on the FC layer’s output via a softmax activation function. Finally, we use cross-entropy loss to quantify our network’s performance and subsequently update the model parameters.

### 2.4. Experiments and Evaluation

#### 2.4.1. Implementation Details

We perform all experiments with TensorFlow 2.4 on an NVIDIA Titan Xp, except for batch sizes of more than 64 total images, where we use an additional NVIDIA Titan X. The CNN architecture uses a batch size of 72 images. We train the LSTM with a batch size of 8 sequences, each sequence holding 9 images, which corresponds to 72 images per iteration. Increasing the batch size in either of the above architectures yields negligible improvements. Nevertheless, using larger batch sizes for the BLSTM shows more significant improvements. Therefore, in the end, the BLSTM is trained with a batch size of 16 sequences, each holding 9 images, for a total of 144 images per batch. It takes approximately 25 min to train the model. This corresponds to 4500 iterations. It iterates through 4500 × batch_size sequences in total. This corresponds to about 16 epochs for the BLSTM and 8 for the CNN and LSTM architectures. Stochastic gradient descent without momentum is used to train the network. In terms of data augmentation, we use colour jittering, which includes adjustments to brightness, contrast, saturation, and hue, random gray-scale conversion, rotations, random left-right as well as up-down flips, and finally introduce a mask with a radius of 128px to remove any possible artefacts near the borders.

#### 2.4.2. Evaluation Strategies

We experimentally evaluate the classification performance of the systems using quantitative and qualitative evaluation strategies. We report evaluation metrics on the test set to illustrate the performance and compare the networks to one another. To date, accuracy is the most frequently used metric for polyp detection. However, it is often not the right choice for data suffering from a class imbalance, since it is highly affected by the predominant class, represented mainly by images without polyps. Moreover, it does not necessarily correlate with the time needed by a physician to make a diagnosis [2]. Hence, we evaluate the systems’ performances using the Area under the Receiver Operating Characteristic Curve (ROC AUC) [23]. The ROC shows the True Positive Rate (TPR) and False Positive Rate (FPR) at different classification thresholds and represents the trade-off between these two indicators given a particular threshold value. The TPR represents the proportion of the positive class that got correctly classified, whereas the FPR stands for the proportion of the negative class that got incorrectly classified. The AUC can be interpreted as the classifier’s probability of distinguishing between the positive and negative class for all existing thresholds. We will refer to the ROC AUC simply with AUC from now on.

Since the system’s overall goal is to minimise the time needed by the physician to analyse the images [2], we also report the recall (sensitivity) at a specificity of 80%,90%, and 95%. The respective recall represents the number of images a physician has to inspect in order to obtain a specific performance. For instance, the recall at a specificity of 80% measures the percentage of detected polyps if only 20% of all images are reviewed [2].

Furthermore, we perform a 5-fold cross-validation [24] to train the systems multiple times with different dataset configurations, increase our confidence in the networks, and select the best-performing model. When using a 5-fold cross-validation, the initial dataset gets split into five approximately equally sized sub-datasets. These five splits are selected so that images from a same patient are all found in the same split. The training and evaluation process is then performed 5 times. Every time, a different split out of the five is selected as the test set, and the remaining 4 sets are used for training. Thanks to the previous selection criteria, we can ensure that the train and test sets do not contain images from the same patient, which would skew the results. Each fold is executed, and the results received from their evaluation are used to produce a mean and standard deviation score. By doing so, we can ensure that we select the best-performing network without running into the issue of just selecting a lucky train-test-split. Moreover, it provides guarantees towards the generalisation capabilities of the model. Additionally, we must ensure that the training and test sets are disjoint, which means they do not have any shared images or procedures. Therefore, we split the data based on the video IDs. To reproduce and compare the results, we decide not to shuffle the data to always result in the same split.

#### 2.4.3. Qualitative Strategies

When using Machine Learning, it is crucial to guarantee that the system makes meaningful decisions. This becomes even more important when working with medical data. To ensure that our model makes reasonable predictions and better understand the system, besides the quantitative techniques, we add a qualitative evaluation to see where and why the model is failing. Ensuring that it can predict more non-complex cases and only fails if the image was hard to classify is decisive for the final performance of our system. In this regard, we validate the best and worst predictions. To do so, we analyse where the network fails and where it predicts correctly.

## 3. Results

### 3.1. Quantitative Results

The classification metric (AUC) for the baseline model (SSL CNN) and the Sequential Models are shown in Table 2. All three networks improve the performance of the SSL network from Pascual et al. [19] on our dataset, showing an increase of 5.31%, 5.33% and 6.43% in AUC, respectively. While the CNN and LSTM architectures show comparable results, the BLSTM outperforms them. To our understanding, this is because by using a combination of the SSL and a sequential architecture, we can extract the spatial features and consider the data’s temporal nature, which increases the amount of information available to the system. The BLSTM’s additional backward pass further increases that amount. For instance, Figure 4 shows that the BLSTM network outperforms the rest of the models by classifying more true positives at a lower false positives count. We receive a final average AUC of 93.83% using the SSL BLSTM network. This is most likely achieved since the bidirectional LSTM provides the network with the highest amount of information as described in Section 2.3. By processing information simultaneously in the forward and the backward direction, the network can receive information from the past (earlier in the sequence) and the future (later in the sequence) at any given time step. This usually results in a better contextual understanding of the network.

We find that the bidirectional architecture scores the highest sensitivity for given specificity rates out of all networks. This concludes that compared to the remaining networks, the BLSTM minimises the time needed by experts to analyse the video content. For instance, when looking at only 20% of all images, the physician would be able to detect 93% of all polyps on average (Table 2, Sens@Spec80 (%), row 4). This compares well to a typical inter-observer variance of 10–20% when endoscopy videos are analysed with minimal computer support.

### 3.2. Qualitative Results

To further explore the performance of the BLSTM, we look at images and their ground truth and predicted labels to analyse whether the system makes meaningful decisions. Figure 5 shows six sequences sampled from various folds of the test set. We can see that the network rarely fails. Most importantly, we have to point out that the number of missed polyps (dashed red circles) is very low. The system performs quite well, even for rather hard to classify images (sequences 2–4). Furthermore, the incorrectly predicted frames are located at the start or end of the sequence, showing the best performance in the middle of the sequences. For mid frames, the network has more contextual information available than for the ones at the start or end of the sequence, making them easier to classify. However, what is even more interesting to see is that if a sequence shows images from both classes, the misclassified frame falls on an image located at the border of the class transition. Meaning the network fails to correctly predict one of the images where the ground truth switches. This phenomenon is present in all selected sequences and is most likely since the BLSTM receives information from both sides and, therefore, might fail in those class transition points. Nevertheless, it always only fails in one frame and then again correctly predicts the following one. Again, this is likely a result of the forward and backward pass of the system.

## 4. Discussion

In the introduction, we raised the question of whether Sequential Models improve a conventional CNN’s performance in WCE polyp detection. To that end, we proposed to add a BLSTM to the SSL network from Pascual et al. [19]. This allowed us to take into account the temporal information of our data. We compared the network’s performance to the SSL approach from Pascual et al. [19] on our dataset, as well as to the performances of an SSL CNN and a SSL LSTM network. Therefore, we used a subset of WCE data that we received from procedures using the Medtronic PillCam COLON2. Our experiments showed that all networks could outperform the SSL architecture. Moreover, the SSL BLSTM substantially improved the performance. It gave better results than all remaining networks in terms of AUC. To our understanding, this is due to the additional backward pass of the system, which makes the system superior in extracting the most relevant temporal characteristics for a sequence of frames. In addition, the resulting sensitivity rates for given specificity rates were encouraging. The qualitative results of the SSL BLSTM were also promising. The number of missed polyps was low compared to manual endoscopy video analysis, and the network seemed to only fail in reasonable situations.

Limitations of this study merely concern the ones of WCE, meaning having the necessary resources. These include the computational resources to run a Machine Learning model, the necessary medical staff and the required camera capsule.

To summarise, using batches of sequences instead of batches of individual images and extracting their temporal information using Sequential Models enriched the available contextual information and improved the performance. This enhancement can positively impact the time spent by physicians on analysing WCE videos.

## 5. Conclusions

Even though computer-aided networks for endoluminal image classification are still in an early research phase, their performance is of great importance. Aiming at reducing the time a physician spends on analysing the videos, a good classification and, therefore, a good pre-selection is essential. Prospective, taking the data’s temporal information into account can significantly improve future methods, as strongly evidenced by our method. The results obtained are a positive step towards developing robust and accurate polyp detection models.

## Figures and Tables

**Figure 1 diagnostics-12-00501-f001:**
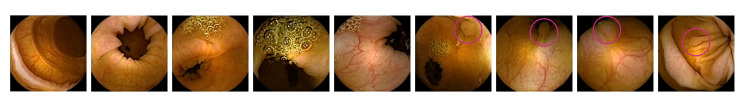
Five random images without polyp, and four images with a polyp obtained from the same patient. As can be seen, the images show high intra-class variability and small inter-class variance. Hence, they can be challenging for the system to classify.

**Figure 2 diagnostics-12-00501-f002:**
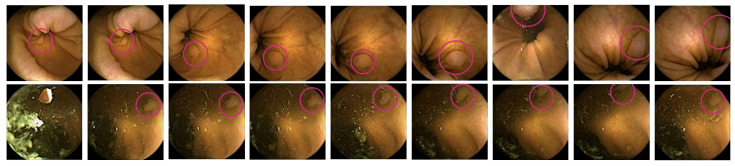
Two sequences in which the physicians have identified polyps. The upper sequence shows the polyp appearing in different locations of the images, whereas in the second sequence, the polyp (upper right corner) stays in approximately the same place while the capsule is moving.

**Figure 3 diagnostics-12-00501-f003:**
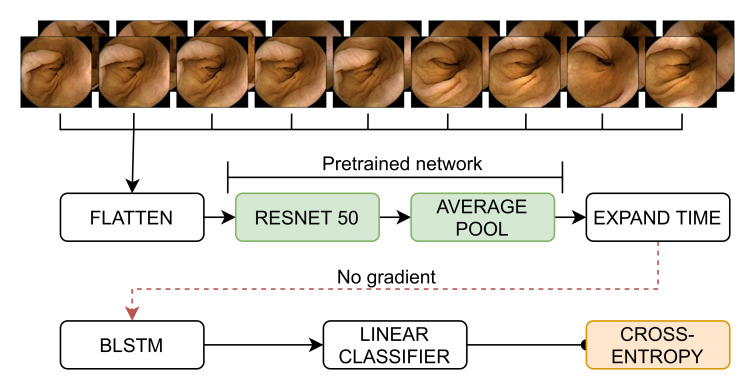
Architecture of the proposed method. Here BLSTM is a Bidirectional Long Short-Term Memory layer. The FLATTEN block removes the temporal axis, flattening over the batch axis, as required per the convolutional blocks in the RESNET 50 block. Likewise, EXPAND TIME recovers the temporal axis for the BLSTM block. The LINEAR CLASSIFIER is a dense layer without any activation, and its outputs are fed into the CROSS-ENTROPY loss (in yellow). The dashed red line denotes that the errors propagated from the classification loss do not change the embeddings learned by the pre-trained network (in green).

**Figure 4 diagnostics-12-00501-f004:**
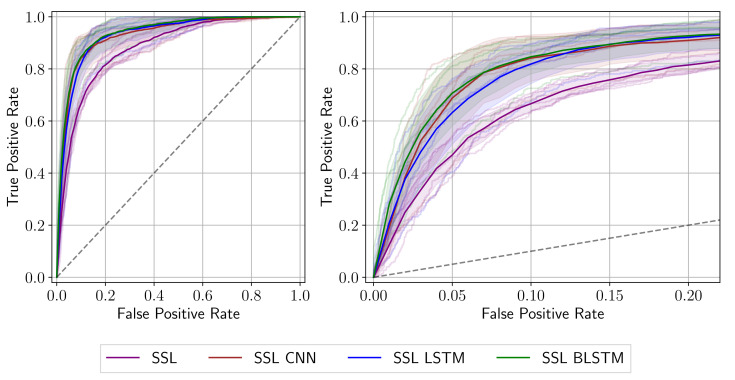
Receiver Operating Characteristic curve of the models studied. Each fold of the cross-validation split is shown in a softer version of its respective model color. The strong line represents the mean of those executions, while the background shade is the standard deviation. A magnified version is provided on the right.

**Figure 5 diagnostics-12-00501-f005:**
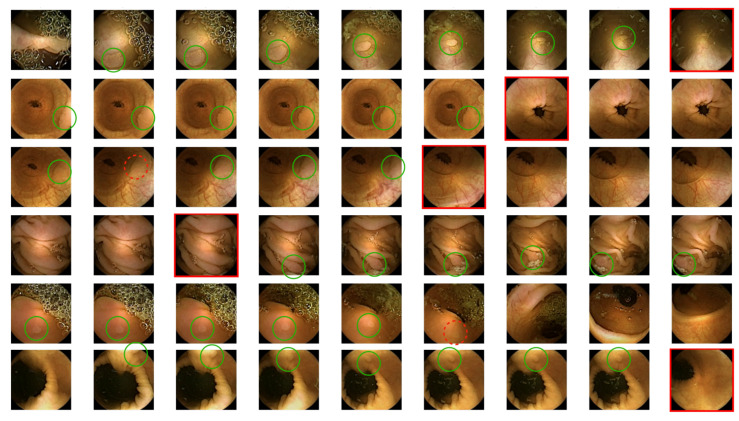
Image sequences. True positive detections (images correctly classified) are marked with a green circle, dashed red circle for False negatives (polyps classified as non-polyps) and images with a red border are False positives detections (non-polyps classified as polyps).

**Table 1 diagnostics-12-00501-t001:** Dataset structure and class distribution. Columns starting with Fold indicate the 5 different folds used during evaluation, as proposed in Laiz et al. [2], with their corresponding Train and Test distribution. Here, Seq. is an abbreviation for Sequences.

	Total	Fold 1		Fold 2		Fold 3		Fold 4		Fold 5	
		**Train**	**Test**	**Train**	**Test**	**Train**	**Test**	**Train**	**Test**	**Train**	**Test**
Videos	110	86	24	87	23	91	19	88	22	88	22
Sequences	5523	4326	1197	4361	1162	4569	954	4417	1106	4419	1104
Images	49,707	38,934	10,773	39,249	10,458	41,121	8586	39,753	9954	39,771	9936
Frames with polyps	2100	1772	328	1521	579	1733	367	1571	529	1803	297
Frames without polyps	47,607	37,162	10,445	37,728	9879	39,388	8219	38,182	9425	37,968	9639
Seq. with solely polyps	105	87	18	73	32	89	16	79	26	92	13
Seq. with at least one polyp	365	312	53	263	102	295	70	276	89	314	51

**Table 2 diagnostics-12-00501-t002:** Mean and standard deviation of ROC AUC and Sensitivity (at given Specificity rates). All metrics have been obtained using a 5-fold cross validation. The best performing system is highlighted in bold.

Network	AUC (%)	Sens@Spec80 (%)	Sens@Spec90 (%)	Sens@Spec95 (%)
SSL	88.16±1.83	81.40±3.38	66.16±3.38	42.85±7.55
SSL CNN	92.84±3.34	90.96±5.96	84.04±8.84	65.62±16.65
SSL LSTM	92.86±3.10	91.94±5.47	81.78±7.70	62.79±11.94
**SSL BLSTM**	93.83±2.65	92.69±5.43	84.44±6.59	70.23±12.62

## Data Availability

Not available.

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
