# Peer review of "Sequential Models for Endoluminal Image Classification"

_diagnostics, 2022, doi:10.3390/diagnostics12020501_

Round 1
Reviewer 1 Report
This is a very interesting work of artificial intelligence recognition of colonic polyps in endoscopic colon video capsules with the Medtronic capsule. I am not qualified as a clinician to validate the mathematical and algorithmic procedure. The results seem to show correct AUC, to be improved. The article is therefore interesting to be published from the angle of polyp detection. A downside: clinical gastroenterologists are not mentioned (for example from the excellent gastroenterology team in Barcelona), although it seems that they were associated with the work of selection and validation of capsule images for computer learning. In this register, we do not see how nurses, for the moment, have a role to play in this stage.
Author Response
Comment 1: Clinical gastroenterologists are not mentioned (for example from the excellent gastroenterology team in Barcelona), although it seems that they were associated with the work of selection and validation of capsule images for computer learning. In this register, we do not see how nurses, for the moment, have a role to play in this stage.
Response 1: Many thanks for reviewing our manuscript and providing feedback. In regard to your comment, we are still collaborating with Dr. Malagelada and Dr. Azpiroz, but in this particular work the dataset comes from another source and there was no implication from them. In this work, we use retrospective Wireless Capsule Endoscopy (Medtronic PillCam COLON2) data from 110 patients that were conducted on behalf of the NHS Highland Raigmore Hospital in Inverness. The obtained data was already labeled by nurses and doctors.
Reviewer 2 Report
- In Fig. 1, it is not clear which image has a polyp and which image has not.
- Section 2.1: The authors removed more “Easy to classify” samples compared to “Hard to classify” ones for fine tune the network, which might lead to biased network. The network may fail to identify simple but frequent samples. The authors should mention the number of samples above for clarification.
- Page 05: Why the average pooling used in stead of the others (i.e., max pooling) is not clear.
- There are no details about the architecture of LSTM and BLSTM. How many layers in those two networks, what was the optimizer, what was the activation function for the internal layer? Is dropout layer or regularization function needed to prevent overfitting?
- Line 251, “For instance, when looking at only 20% of all images, the physician would be able to detect 93% of all polyps on average.” It needs a reference.
- The source of the dataset is not mentioned clearly.
- It is necessary to provide the details of training and testing datasets including how many images were used, etc. Without these, the experimental results are not very convincing.
- True Positive Rate and False Positive Rate are not defined clearly.
- If the numbers in Table 2 came from the image level, it is necessary to provide how many polyp and non-polyp images were used.
- The indentation of Line 155 is not correct.
- Figure?? in Line 164.
Reviewer 3 Report
The authors compared the performance of using Sequential Models for fine tuning with using solely networks that extract only spatial information. They showed that, by taking into account the temporal nature of the data, the detection of polyps in Wireless Capsule Endoscopy videos can be improved.
I read the study with interest. This is a well-designed and written study with clear methodology and interpretation of the results. Discussion is poor and needs to be improved. I sincerely congratulate the authors on their excellent work.
I have few several suggestions for improvement:
- Abstract should be redesigned and results should be presented in abstract. This descriptive type of abstract with repetition of well-known facts from literature is not well designed. Please revise.
- The authors stated that images were analyzed by multiple physicians and expert nurses and tagged accordingly to whether the experts had identified a polyp in an image or not. Please clearly indicate which number of physicians and expert nurses evaluated images and what their level of experience was.
- Please indicate exact inclusion / exclusion criteria. Which patients were included in the study?
- Please indicate primary and secondary outcomes of the study in methodology.
- Line 164 – Please replace ‘Figure ??’ with correct figure number
- Figure 3 – Please provide legend with description of abbreviations used in this Figure.
- Please provide the full title for abbreviation ROC AUC, as it was not mentioned in text.
- Discussion is very short and poor. Some parts from introduction should be moved to discussion (mostly whole chapter 1.2) Discussion section needs to be re-written/re-arranged. Do not present a review of literature in this section. Do not discuss your findings piecemeal. Focus on results from the main objectives of the study. Write in four sequential paragraphs (without headings); (i) summary (not data) of findings from present study; (ii) logical and coherent comparison with existing literature with focus of comparison on main objective(s); (iii) limitations of the study; and (iv) Implications for practice/policy/research with a concluding statement
- Limitations of the study have not even been mentioned.
- Please provide conclusions in separate paragraph.
Round 2
Reviewer 2 Report
The figure number needs to be fixed in the following (Line 202):
"Furthermore, we test the pre-trained SSL architecture on our dataset.
202 Figure ?? shows the final architecture of the SSL BLSTM network."
Author Response
Comment 1:The figure number needs to be fixed in the following (Line 202):
"Furthermore, we test the pre-trained SSL architecture on our dataset.
202 Figure ?? shows the final architecture of the SSL BLSTM network."
Response 1: Many thanks for pointing that out again. We adjusted it and it is now correctly referenced (Figure 3).
Reviewer 3 Report
The authors revised the manuscript according to reviewers comments. The manuscript is significantly improved after revision.
The following changes are needed before favourable decision should be made:
1. In my opinion presentation of the results and discussion should be separated. The authors should first present their results and in the next paragraph discussion which should be extended.
2. Please provide the ROC diagram in your results.
Author Response
Comment 1: In my opinion presentation of the results and discussion should be separated. The authors should first present their results and in the next paragraph discussion which should be extended.Response 1: Many thanks for the additional comments on our manuscript. We have split the Results and Discussion section again into two separate chapters. When doing so, we tried to adapt to the general structure of Machine Learning papers and the Special Issue's one. Therefore, we renamed the former Conclusions chapter to "Discussion" and added a new, short Conclusion, as seen in other papers from our Special Issue. We hope this meets your expectations.
Comment 2: Please provide the ROC diagram in your results.
Response 1: Many thanks for the suggestion. We have added the diagram of the ROC to chapter 3 (Results) and we are also referring to it in the text.